# Design of a Forest Fire Early Alert System through a Deep 3D-CNN Structure and a WRF-CNN Bias Correction

**DOI:** 10.3390/s22228790

**Published:** 2022-11-14

**Authors:** Alejandro Casallas, Camila Jiménez-Saenz, Victor Torres, Miguel Quirama-Aguilar, Augusto Lizcano, Ellie Anne Lopez-Barrera, Camilo Ferro, Nathalia Celis, Ricardo Arenas

**Affiliations:** 1Escuela de Ciencias Exactas e Ingeniería, Universidad Sergio Arboleda, Bogotá 11011, Colombia; 2Earth System Physics, Abdus Salam International Centre for Theoretical Physics, 34151 Trieste, Italy; 3Facultad de Estudios Ambientales y Rurales, Pontificia Universidad Javeriana, Bogotá 11011, Colombia; 4Instituto de Estudios y Servicios Ambientales-IDEASA, Universidad Sergio Arboleda, Bogotá 11011, Colombia; 5Departamento de Ingeniería, Aqualogs SAS, Bogotá 11011, Colombia; 6Dipartimento di Ingegneria Civile, Edile e Ambientale, Università degli Studi di Padova, 35122 Padova, Italy; 7Departamento de Medio Ambiente y Sostenibilidad, Universidad Andina Simón Bolivar, Sucre 703030, Bolivia; 8Centro de Investigación de Filosofía y Derecho, Universidad Externado de Colombia, Bogotá 11011, Colombia

**Keywords:** wildfires, early alert system, machine learning, bias correction, Naïve-Bayes classification, susceptibility, risk assessment, soil protection

## Abstract

Throughout the years, wildfires have negatively impacted ecological systems and urban areas. Hence, reinforcing territorial risk management strategies against wildfires is essential. In this study, we built an early alert system (EAS) with two different Machine Learning (ML) techniques to calculate the meteorological conditions of two Colombian areas: (i) A 3D convolutional neural net capable of learning from satellite data and (ii) a convolutional network to bias-correct the Weather Research and Forecasting (WRF) model output. The results were used to quantify the daily Fire Weather Index and were coupled with the outcomes from a land cover analysis conducted through a Naïve-Bayes classifier to estimate the probability of wildfire occurrence. These results, combined with an assessment of global vulnerability in both locations, allow the construction of daily risk maps in both areas. On the other hand, a set of short-term preventive and corrective measures were suggested to public authorities to implement, after an early alert prediction of a possible future wildfire. Finally, Soil Management Practices are proposed to tackle the medium- and long-term causes of wildfire development, with the aim of reducing vulnerability and promoting soil protection. In conclusion, this paper creates an EAS for wildfires, based on novel ML techniques and risk maps.

## 1. Introduction

Although fire is a natural mechanism of the self-regulation of terrestrial ecosystems, anthropic activities may alter natural regimes, leading to a decrease in their capacity to reduce the intensity, magnitude and spread of forest fires [1]. Moreover, frequency and intensity changes in wildfire patterns contribute to increased rates of Greenhouse Gas (GHG) emissions, a loss of land cover, an acceleration of erosive processes and a higher pressure on biodiversity [2]. This situation has entailed high costs in terms of capital and human resources, which have been mainly directed towards environmental reparation, due to the lack of adequate protocols and methods for an effective prevention. 

In *Colombia*, wildfire events are highly correlated with the spatial distribution of the territories [3]. These mainly occur during the dry season and are strongly related to the expansion of agriculture and livestock frontiers [4]. During 2020, the number of forest fires in *Colombia* increased by 32% compared to 2019 [5]. Likewise, it is important to note that the most populated regions, the *Andean*, *Orinoquia* and *Caribbean*, are also the most exposed to wildfire hazards. Some studies have demonstrated the high vulnerability of Colombian ecosystems to wildfires, an issue expected to exacerbate due to climate change (due to changes in temperature and humidity, especially at the North of the country [6] and human intervention on territories [3]). In this context, a systemic change is required to transform the country’s approach to wildfire risk. According to Thompson et al. [7], authorities should, among other recommendations, ‘prioritize investments based on who can most efficiently mitigate risks and ensure that decisions are made based on relevant information’. Thus, formulating methodologies that utilize available and relevant information, which, in turn, can be used by stakeholders in the territories, may reduce the impact of wildfires in *Colombia*. In this sense, risk assessment is a critical part of forest fire prevention, as it fosters planning through objective tools aimed at controlling the negative effects of wildfires [8].

Wildfire risk mitigation, and the correlative assessment of hazard and vulnerability, should be considered under a particular framework, due to its specific features. Unlike other types of hazards, a wildfire is a multicausal event. For a wildfire to occur, certain conditions related to the combustible biomass, the ignition event and the spreading of fire must be met. The conditions concerning biomass and fire spreading are related to meteorological, geological and land cover variables, while ignition is a random event, in most cases, related to a direct human intervention [9]. On the other hand, the evaluation of vulnerability, in turn, must consider ecological and socioeconomic aspects to adequately assess the risk and allow for appropriate decision making [10]. Hence, the characterization of fire vulnerability to wildfires requires an assessment of exposure, sensitivity and resilience [11].

Understanding the importance of supplying territorial decision makers with relevant and useful information, this paper proposes a wildfire risk management plan that contains: a tested methodology to spatially predict the daily probability of wildfire occurrence 24 h prior; a strategy to measure global vulnerability towards wildfires; and their integration into risk maps. This plan was used in two Colombian regions: *Cundinamarca* and *Magdalena*. The first part of this paper focuses on the prediction of the probability of the occurrence of a wildfire in two case study areas, considering past ignition events. Likewise, the term “wildfire occurrence” regard favorable conditions related to biomass and fire spreading. Two models are used, and the results are jointly presented: first, a neural network quantitatively estimates the probability of a wildfire occurring using meteorological data; second, a Naïve-Bayes model estimates the probability of wildfire occurrence considering Land Cover Type, vegetation density and moisture conditions in vegetation soil. As a result, the output of this model to predict wildfire can be used as an early alert system.

We decided to use ML models, which especially relate to convolution neural networks (CNN), due to the strong results they have shown in multiple science fields. The convolutional networks have been mainly used for visual processing. For example, Yu et al. [12] created a state-of-the-art CNN to identify cracks that develop on concrete structures. The precision of their networks was greater than 90%, which shows their capabilities in civil engineering, and their potential in many other fields and applications. One of those applications is shown in Javanmardi et al. [13]. They created a method which reduces the limitations of CNNs when captioning images. This has increased the attention on the advantages of CNNs when identifying and classifying objects, and it also sheds light regarding the capability of the network when forecasting. One interesting example relates to air quality, which has been diversely studied. For example, Kalajdjieski et al. [14] show that CNNs are able to predict the amount of pollution from satellite images, without using raw time series. This is certainly of great utility for places without ground stations. Another example is the work of Kabir et al. [15], which also used satellite images to determine the spatial distribution of pollution. These works, and many others, some of which are referenced in this paper, have built the basis for the application of CNNs to different environmental problems, which can be solved with satellite images and/or time-series of data, as the first part of this paper shows.

The second part presents an assessment of global vulnerability, which integrates the synergy between two types of vulnerability: ecological vulnerability, which evaluates the ‘natural’ land (hereafter land cover) potentially affected by wildfires, and social vulnerability, which uses socioeconomic aspects to define human response capacity to a wildfire and, in other terms, measures the level of social impact. The third part integrates the analysis of the probability of occurrence and vulnerability to produce risk maps. These results will facilitate the identification of critical spots where preventive actions should be focused. Finally, a set of preventive and corrective measures is proposed. These measures are designed to reduce the ecological vulnerability and to promote soil protection. In these terms, the results can aid the process of decision making of environmental risk management authorities, concerning prevention, mitigation and recovery planning, empowering response capacity and generating territorial resilience.

In general, this paper presents the design of an early alert system, with some novel approaches explained below: (i) We used a state-of-the-art bias-correction method for meteorology, based on ML techniques, which, to the extent of our knowledge, is one of the first to be applied in an area with complex terrain (i.e., *Colombia*). (ii) This is one of the first efforts to use a Navier Bayes classifier to determine the probability of the occurrence of a wildfire, an approach validated with several methods from the literature. (iii) We merged this approach with a vulnerability analysis in risk maps, which allows for the dynamic assessment of the risk and the daily changes. Likewise, it provides insights on how and where to proceed to reduce the global risk. These insights are merged into a new protocol, which includes short- and long-term actions. (iv) Today, the access to, and development of, information in *Colombia* is still mostly centralized, with some national governmental agencies working on generating maps and tools that aid decision making in the territories on coarse space–time scales. Providing an easy-to-use tool, with dynamic results at appropriate spatial scales, contributes to strengthening territorial (communitarian) sovereignty by handing control and promoting knowledge and skills to local levels.

## 2. Materials and Methods

The proposed method integrates several elements to calculate wildfire risk. Following the terminology proposed by Bachmann and Allgöwer [9], wildfire risk calculation considers the probability of occurrence and the outcome. As the probability of ignition is highly dependent on human activities, the probability of occurrence in this paper will only examine the probability of fuel conditions allowing a fire to last and spread. Meanwhile, the outcome, defined ‘as the weighted sum of the potential impacts on each affected object’ [9], will be explored through the analysis of what we call global vulnerability. 

In this study, the probability of the occurrence of a fire is expressed as the result of computing two main elements: the Fire Weather Index (FWI) [16], and a proposed index called the Fire Land Cover Index (FLCI). The former, the official index for predicting fire risk in *Canada*, accounts for weather conditions. In contrast, the latter considers land cover features such as density, previous moisture content and Land Cover Type. 

### 2.1. Study Areas

Two study areas in *Colombia* were selected. The first area is *Cundinamarca-Bogotá* (hereafter *Cundinamarca*), located in the center of the country (lon: −74.8906 to −73.0508; lat: 3.7270 to 5.8367). The second, *Magdalena*, is located North (lon: −74.9456 to −73.5420; lat: 8.9171 to 11.3494) (Figure 1). Study areas were chosen because they had a large number of hotspots between 2010 and 2021. Likewise, as the zones have a high population density, social vulnerability can also be included in the analysis. The hotspot information was retrieved from the Fire Information for Resource Management System (FIRMS), measured with the Moderate Resolution Imaging Spectroradiometer (MODIS) [17], and with the Visible Infrared Imaging Radiometer Suite (VIIRS). For this study, a wildfire event was considered if the reliability of the measurement was higher than 80%, and the brightness temperature was higher than 360 K [18]. Nevertheless, 370 K, 350 K and 340 K were also used as thresholds, and delivered the same results. On the one hand, *Magdalena* has a high density of intense hotspots and is the department with the largest area and number of urban centers. Meanwhile, *Cundinamarca* does not have the highest density of hotspots, but has the highest population density and the greatest data availability.

### 2.2. Occurrence Probability of a Wildfire

In order to calculate the probability of the occurrence of a wildfire, two different paths were followed. First, a meteorological forecast model was constructed with two different techniques: (i) Two machine learning (ML) forecast models and (ii) a ML bias correction method. Second, these results were combined with a Naïve-Bayes classifier, which served to calculate the FLCI. For this purpose, two ML approaches were used to predict the meteorology. Both approaches were used considering that one model could deliver better predictions in places with in situ observations, and the other in zones without ground-base stations, which are the cases of *Cundinamarca* and *Magdalena*, respectively.

We evaluated the performance of two models to simulate 4 meteorological variables (2m temperature (T), wind speed, relative humidity (RH) and precipitation). The first approach regarded a 3D convolutional network, with the construction of two models: a 3D convolutional neural network (hereafter 3D-CNN) and a 3D convolutional merged with a Long–Short Term Memory (LSTM) network (hereafter 3D-LSTM). The second approach consisted of a bias correction model based on Sayeed et al. [19,20], where the Weather Research Forecast (WRF) model was coupled with a 1D convolutional layer joined with a bidirectional network (WRF-1D), whose structure was based on Celis et al. [21]. 

#### 2.2.1. Description of the 3D Convolutional Networks and Evaluation

For the construction of both convolutional networks, CAMS Near-Real Time Satellite Data (CNRT) were used with a spatial resolution of 0.4° and a 6-hourly temporal resolution, in the period between 1 January 2017 to 31 December 2021. The input (X-matrix), output (Y-matrix) and structures were based on the work of Huang and Kao [22] and Casallas et al. [23]. The CNRT data had a five-day lag. Hence, the simulation had to forecast 7 days to produce 24 h in the future. For this reason, the X-matrix was constructed using 56 data points (14 days) and the Y-matrix had 28 time-data points (7 days) of the forecasted variable. For example, to forecast temperature, the input was the same variable with a time lag, and the output was a period of time in the future, so no multicollinearity analysis had to be performed [5]. Two network structures were designed to account for the better results that each structure produced depending on the simulated variables (Figure 2 and Appendix A). To reduce the uncertainty to the greatest extent possible, a grid search method (e.g., [24]), which consists of more than 350,000 tests, was implemented to find the best hyperparameters for each network structure. Moreover, an early stopping method was applied to prevent overfitting [25]. It is also important to mention that 6 other network structures were tested, and thereafter we selected the one producing the smallest errors.

With these results, the Fire Weather Index (FWI) [16] was calculated following the approximations of Di Giuseppe et al. [26] for *Colombia*. Thereafter, this index was used to evaluate the model’s performance. Later, the models were validated for 160 days, which were removed from the training set. The data were separated according to whether a fire event occurred (i.e., 50% with fire events and 50% without). With this distribution, we validated the model’s capability of reproducing both scenarios. In addition, every month of the year had at least 5 days of representation (not continuous days) in order to evaluate the different rainy seasons and also a variety of weather conditions (e.g., wind field, precipitation). Additionally, we used 3 statistical and 3 categorical parameters [27] to evaluate the model’s capability to follow trends, represent magnitudes and identify events. These statistical parameters were also used to evaluate the WRF-1D model.

#### 2.2.2. Description of the WRF-1D Model and Evaluation

The nonhydrostatic WRF model version 4.2 [28] was used to forecast meteorological variables. Two nested domains were used for both cities, with 32 vertical levels (model top at 20.5 km) and a horizontal resolution of 9–3 km^2^, respectively, for each domain. The output was 3 hourly and the parameterizations were based on [29,30] (See Appendix A for more details). The model was executed for 4 years (1 January 2017 to 31 December 2021) to produce the training data for *Cundinamarca*. Its initialization and boundary conditions (6 hourly) were calculated from the 0.5° GFS-reanalysis. It is important to mention that the bias correction technique was only applied to *Cundinamarca*, since *Magdalena* did not have in situ stations. Thus, for this area, the model could not be corrected or validated.

Previous studies (e.g., [31]) have shown that multicollinearity can produce large errors in the training of neural networks. Consequently, the Variance Inflation Factor (VIF) was applied, following the work of Ghahremanloo et al. [32]. We selected the variables that produce a multicollinearity smaller than 5, as suggested by Kline [33]. As our purpose was to bias-correct the variables that serve as input for the FWI, we applied the VIF (on the 36 variables from the WRF) 4 times, one for each of the output variables. We found that 10 variables from the WRF output (i.e., relative humidity, 2m temperature, wind components (u and v), precipitation, latent and sensible heat flux, vertical velocity, Outgoing Long-wave Radiation (OLR) and planetary Boundary Layer Height (BLH)) and 4 (i.e., RH, wind speed, precipitation and 2m temperature) from the in-situ stations of *Cundinamarca* were able to avoid multicollinearity and had predictive potential. These selected variables were similar to the ones found in previous papers (e.g., [20,21]), since they had a very strong predictability and did not produce strong collinearity.

The WRF-1D model was trained as follows: The X-matrix contained the 10 WRF variables for the period intended to be simulated (24 h in the future), while the station variables were input in the past (24 h prior). In the case where data need to be imputed, we followed the Mogollon-Sotelo et al. [34] method. The Y-vector consisted of each one of the station variables in the future, so the network had to be trained multiple times to produce a prediction for each one of the stations and variables, and the structure can be found in Figure 3. There were two main reasons behind the use of this structure: (i) we tested 6 different structures with different combinations of layers (e.g., with and without bidirectional networks) and found that this configuration produced the smallest errors; and (ii) the bidirectional networks were able to produce nowcasting when used on a time-series of data [21] due to their ability to reverse-order the data. This helps them identify strong events with more ease than, for example, a normal LSTM [35]. In fact, one of the drawbacks of the LSTM is that in some cases it is not able to reproduce the peaks of the variable they are forecasting (e.g., PM_2.5_), something that is largely reduced with the bidirectional network (see [21,23] for examples). The WRF-1D results were used to calculate the FWI for the locations of the six stations. Thereafter, it was evaluated for 160 days selected for the 3D-CNN and the 3D-LSTM validation. This approach used the same statistical/categorical parameters as the 3D networks. A more detailed description of the models can be found in Appendix A.

#### 2.2.3. Fire Land Cover Index

To predict the FLCI, the Naïve-Bayes classifier (Equation (1)) was implemented as a Bayesian neural network. The classifier calculated the probability of wildfire occurrence (P) as a function of the land cover by assigning the most likely output (Y) for a given data set that contains an N number of inputs ($X_{i}$) [36]. However, it particularly assumes that $X_{i}$ is independent for any given Y.
(1)$$P_{X|Y}=\prod_{i=1}^{N}P_{X_{i}|Y}$$

Land cover features were considered independently of climatic conditions to prevent multicollinearity, as stated in the previous section. The analysis considered, on a daily resolution, three X-matrices: the Normalized Difference Vegetation Index (NDVI), Normalized Difference Water Index (NDWI) and Land Cover Type (LCT) from 1 January 2017 to 31 December 2021. These data were upscaled to reach the FWI spatial resolution using Power Law Averaging [37]. The Y-matrix contained daily registered hotspots represented as zero (no wildfire) and one (wildfire event). Likewise, the matrix had a 1-day positive lag in order for the network to produce a forecast similar to the one of the meteorological models. These data were also upscaled to the FWI resolution using the maximum value as a reducer as in Kriebel et al. [38]. The selection process of the input variables considered results from its documented relation with wildfires, from its affordability [39,40,41], and also from the results of the VIF method described in Section 2.2.2.

Daily X-matrices data were built using imagery from Terra satellite’s MODIS-MOD09A1 version 6.1 sensor (0.5 km resolution), which were preprocessed as described in Appendix A following Ackerman et al. [42]. The spectral indices NDVI and NDWI [43] were recategorized into positive integers from zero to one. Thereafter, the land cover classification was implemented using the algorithm proposed by Simonetti et al. [44], in which positive integers are assigned to 13 categories. Clouds and shades were masked using images from the QABand and Bitmask for QA (Appendix A). Then, the X-matrices and the Y-matrix were used to train the network, and the identified 160 days were removed for the purposes of meteorological validation. To train the data, we used a k-fold cross-validation method with 10 folds. Thereafter, to validate it, the accuracy rating was used on the aforementioned 160 days. As the results of this method are probabilities, we also applied the receiver operating characteristics (ROC) area under the curve method [38]. It is important to mention that this method has not been used before for *Colombia*, so the results were compared with the results of a deterministic first order model (Equation (2)) that incorporates the NDVI, NDWI and LCT to quantify the probability of wildfire occurrence due to land cover and three weights (W), a model that has been validated and used in previous studies (e.g., [9,39,45]).
(2)$$FLCI_{T}=W_{1}NDVI_{T}+W_{2}NDWI_{T}+W_{3}LCT_{T}$$

### 2.3. Global Vulnerability and Wildfire Risk

#### 2.3.1. Global Vulnerability

The global vulnerability assessment was based on Chuvieco et al. [10] together with Vargas-Sanabria and Campos-Vargas [46]. Their methods were used to calculate both social and ecological vulnerability (i.e., overall vulnerability) of both study areas. Information from several studies was filtered using the following criteria: (a) articles that do differentiate ecological and social vulnerability, (b) articles that did not specifically focus on vulnerability to wildfires, (c) articles that do not have the name of journals and/or authors who do not confirm their origin. 

The Information collected was analyzed with a multicriteria matrix, with the aim of evaluating the suitability and the basis of the criteria used by the authors of these articles. With this information, we identified frequently used criteria in other studies and selected some that can be used for this study. Additionally, we performed another filter of selection by examining the data availability of the study area. Thereafter, a double-entry matrix was built, where the conducted review and the identified criteria, within each case study regarding ecological and social vulnerability, were crossed. To recognize the common criteria when crossing the information, each crossing was assigned a value of zero if the criterion was absent for the case study or one if the criterion was available. Subsequently, we added the columns corresponding to each criterion to ascertain which were more frequently used. The summation is presented on a scale between 1 to 13, one representing the least used criterion and thirteen if the criterion was evaluated in every study. The criteria with a score ≥ five (the average of the points of all documents), and also those considered of great importance by the authors, according to the references, were selected.

To analyze global vulnerability to fires, eight criteria were used, four of which related to socioeconomic vulnerability and the rest related to ecological vulnerability. The criteria were evaluated using publicly available secondary information. The ecological vulnerability index takes into account the types of covers according to their adaptation to fire. To determine these types of covers, we used the publicly available map of land cover for the period 2010–2012 [47]. Thereafter, the hedges were identified in accordance with the combustion capacity classification of Paramo-Rocha [48]. Coverage classification is predominantly related to fuel type and gives the susceptibility category described in Table 1.

The second criterion calculated for ecological vulnerability is related to threatened ecosystems. We softened the classification of Etter et al. [49], superimposing it with the identified vegetation covers. Then, values for ecosystem categories were assigned as follows: critically endangered (4), endangered (3), vulnerable (2) and of minor concern (1). We used a method from Miranda et al. [50] to calculate the third criterion of ecological vulnerability, which defined the Wildland–Urban Interface (WUI) as an area that surpassed the threshold of 6.17 dwellings km^−2^, dominated by wild vegetation. Likewise, we adapted 1 km^2^ as a minimum size threshold for heavily vegetated areas, to avoid including residential areas that are within 0.5 km of small urban parks. Additionally, for the zone to be considered a WUI, more than 50% of its area must be covered by wild vegetation. On the other hand, for the interface, the vegetation cover must be less than 50% with fragments of dense forest (>75%). Regarding these criteria, the WUI interface was used as an indicator, adapted from Galiana-Martin et al. [51]. Their methodology analyzed the peripheral areas of urban centers adapting the Corine land cover method. With it, it is possible to differentiate zones that entirely border areas of forestry activity with different forest covers, from those that border areas of crops, riparian forest, etc. This comparison can aid in the assessment of those areas with greater risk within the forest urban interface of the commune. To achieve this, we classified the areas into 3 categories according to the following interface: 0: wet areas; 1: urban area, transitory crops, artificial green areas; 2: heterogeneous agricultural area, open areas without little vegetation, permanent crops, agricultural areas; and 3: herbaceous vegetation, pastures, forests, shrubs.

To assess socioeconomic vulnerability, the WUI determinants for ecological vulnerability were used as an indicator proximity criterion of WUI related to urban areas. These criteria were measured by WUI number near urban centers less than 1 km away [52]. To determine the urban centers, we used the information available in the population density map [53]. The last criterion relates to the local response capacity, and the data were recovered from the Departamento Nacional de Planeación-DNP [54]. We used the capacity index according to their financial, socioeconomic and institutional capacity, since these dimensions define the actions that public authorities can take in the face of disaster risk management.

Weighting the global vulnerability criteria entailed assigning weights according to the multicriteria evaluation. Based on this analysis, six criteria were selected, three related to ecological vulnerability (covers and adaptation to fires, strategic ecosystems and the Wildland–Urban Interface (WUI)) and the other three related to social vulnerability (the occupation of the Wildland–Urban Interface (WUI), adjacent area and response capacity). The defined criteria were weighted according to the valuation assigned in the criteria selection matrix (see Appendix A). Then, the percentage value was assigned to each variable, obtaining Equation (3) to calculate total global vulnerability.
(3)$$V=0.25WUI_{i}+0.18C_{i}+0.12CR_{i}+0.17TC_{i}+0.08EC_{i}+0.2WUI_{ei}$$
where V is the global vulnerability, $WUI_{i}$ represents occupation in the WUI, $C_{i}$ is the adjacent area, $CR_{i}$ is the response capacity, $TC_{i}$ represents the types of covers (adaptation to fire), $EC_{i}$ are the strategic ecosystems and the $WUI_{ei}$ indicates the Wildland–Urban Interface (WUI). These criteria were integrated with the aim of obtaining five associated categories of vulnerability, i.e., very low, low, moderate, high and very high/extreme.

#### 2.3.2. Probability of Wildfire Occurrence

With calculations of the FWI (meteorology) and the FLCI (land cover), it is possible to calculate the probability of wildfire occurrence by using Equation (4).
(4)$$WFP_{t}=W_{1}FLCI_{t}+W_{2}FWI_{t}$$
where WFP is the probability of wildfire occurrence for a time t, and $W_{1}$ and $W_{2}$ are weights assigned 0.5 each. The FLCI and FWI were previously normalized (Table 2). As the importance of climatic and land cover variables, as predictor variables, fluctuates among different studies (e.g., [40,41,55]), equal weights were assigned, with the aim of reducing the bias. The WFP serves as an estimate of the probability of wildfire occurrence. This probability is ranked in five categories: extreme, very high, moderate, low and very low danger [41].

#### 2.3.3. Wildfire Risk Management

We considered wildfire risk as the mathematical relationship between the probability of wildfire occurrence and the expected outcome for a particular time period in a particular geographical area. This risk was quantified with Equation (5) as in [56]:(5)$$FR_{t}=WFP_{t}V_{t}$$
where $FR_{t}$ is the probability of wildfire risk for a time period, and $WFP_{t}$ and $V_{t}$ are the normalized values for the probability of occurrence and global vulnerability for a particular time period, respectively. Finally, the probability of wildfire risk was classified into five ranges. This process considered the individual documented indexes of fire sensitivity classification. Table 2 presents the normalized classification for each index and the resulting classification for wildfire risk.

The design of the protocol for the EAS was divided into two parts: the first comprises the management measures that should be carried out by the communities, government bodies and associations when an alert or risk of forest fire is identified in a certain area, to prevent a wildfire event or to reduce its impacts. The second part relates medium- or long-term actions to reduce the risk of ignition and the environmental impacts associated with the soil, and to improve the edaphic conditions of the area at risk (before or after the ignition).

For the first part, using the calculated risk (on a spatiotemporal scale) and the stoplight-type measurement scale, a conceptual frame of reference of the EAS for the forest fires was designed, which has the potential to be adapted to the regulatory, institutional and infrastructure conditions of any country. This framework can aid in the identification of critical points, to focus efforts on preventing the generation and spread of fires (e.g., soil management policies). 

The integration of risk variables to determine fire areas generates key information when developing institutional response protocols, which must include risk management actions from government agencies, social organizations and territorial actors. Fire risk management can focus on four areas: 1. combating fires in protected areas; 2. combating fires in other areas; 3. preventive actions; and 4. support for socioenvironmental actions [57]. In addition, fire management should focus on combining studies and research regarding fire as an ecological factor, investigating its origin and causes, and considering the use of preventive, suppressive or fire-fighting elements. Its scope can be national, regional or related to protected areas at high risk of forest fires [58]. In areas of potentially increasing fire risk, selecting safe and effective fire control tools is key [59]. These tools are essential to prevent and manage fires when they are timely and readily available, as they increase the response capacity of vulnerable communities concerning risk management and wildfire response planning. Specifically, preventive actions related to forest fires can be integrated into the second part of the protocol to manage areas and risk.

For the second part, we proposed Soil Management Practices (SMP) before and after a wildfire event is identified. These SMP practices seek to prepare the soil to make it more resistant and to ensure a better recovery of its properties. These practices were developed in two phases: (i) the generation of preliminary pre- and postfire Soil Management Practices and (ii) the creation of integrated recommendations adapted to each implementation zone.

**Phase 1—Design of sustainable Soil Management Practices:** to determine preliminary management practices, public policy and management documents (15) from different Latin American countries (*Mexico*, *Colombia*, *Peru* and *Chile*) were evaluated (see Appendix A), taking into account their potential for replicability in the region. The evaluation consisted of a double-entry matrix where the presence of the practice was valued one and, conversely, zero if not present. Subsequently, the values were aggregated to prioritize the inclusion of the practices in the SMP protocol. It is important to note that the practices were divided into pre- (8) and postfire (11) practices.**Phase 2—Creation of integrated recommendations:** The SMP actions previously identified were redesigned to include the physical characteristics of the territory, such as: land use (to articulate recommendations related to the activities conducted in the zone), land cover, the susceptibility of the vegetation cover to ignition obtained from [48] and the change in the susceptibility of the cover to ignition over time. It is important to mention that in this study, the moisture content was not included as a physical parameter for two reasons: (i) the IDEAM [60] method allows the calculation of susceptibility without considering soil humidity, and without increasing uncertainties; and (ii) neither area has data for the soil moisture content. Despite the existence of reanalysis products for this variable, without in situ measurements to validate them, it is better to avoid its usage.

## 3. Results and Discussion

### 3.1. Meteorological Model Evaluation

To determine the precision of the models, the Root Mean Square Error (RMSE), the Index of Agreement (IOA) and the Correlation Coefficient (Rho) were calculated as statistical parameters, and the Hit Rate (HIT), the False Alarm Rate (FAR) and the Proportion of Correctness (POC) were used as categorical parameters. Likewise, the threshold of the latter was set to 28 based on Varela et al. [61].

#### 3.1.1. 3D-Convulutional Networks Evaluation

Before describing the model’s results, to reduce the uncertainty, we calculated the forecast horizon following the work of Stergiou and Karakasidis [35] (consult the reference for methodological details). The horizon was 8.33 days, and the system was found to be chaotic. This was to be expected since Lorenz [62], in his seminal paper, showed the importance of chaotic systems, and used the meteorological system as an example. Moreover, we performed a battery of experiments (on the 3D networks) where we changed the forecasting horizon from 5 to 10 days. We found that, after 9 days, the errors started to sharply increase.

Figure 4 and Appendix A (see Appendix A) shows the model evaluation parameters’ average of the 160 days previously selected (see Section 2.2.1 for details). The model delivered good results in terms of precision for both selected areas. In most regions, Rho was larger than 0.5, while the lowest values were in the mountain ranges. In terms of its magnitude, the RMSE showed values smaller than five in more than 80% of the domain. The IOA had values larger than 0.5 and more than 60% of the domain. These results, when compared with the model benchmarks of Guevara-Luna et al. [30], indicate that the model performed well when reproducing the magnitude and the behavior of the FWI in the region. In terms of categorical metrics, for *Cundinamarca*, the model identified more than 80% of the events inside the department and did not produce false alarms, except for its central area, possibly due to topography uncertainties and human interactions. In *Magdalena*, on the other hand, it produced less than 5% of false alarms in most of the domain. This was complemented with high values in the HIT parameter, larger than 90% in the north of the territory, in contrast to the low values in the Caribbean Sea and in the center of the territory, where only 40% of the events were identified. This error could be explained by the region having large uncertainties in terms of wind speed, caused by the changes produced by its highly complex topography.

To have a more robust evaluation of the 3D models, we compared our results with previous studies. For example, Anderson et al. [63] developed an alert system for protected areas of South America. They calculated the HIT and FAR parameters and found that their model had good precision. However, the model presented herein had a 40% higher accuracy in the HIT category and produced 5% less false alarms. On the other hand, Di Giuseppe et al. [26] used a forecast model to calculate the potential predictability of wildfires around the globe. They found a categorical score very close to ours. In fact, their score oscillated between 0.6 to 0.9, similar to the values in Figure 4. Nevertheless, they forecasted the entire world, and not a specific zone. Hence, although the comparison was possible, it is important to bear in mind that this paper focuses on only one zone. Hence, their precision could increase more. These two comparisons show that 3D networks have a great accuracy when compared with other models and methods, and that, in contrast to other works, this model was combined with a land cover analysis, which improved the results. 

In summary, the model was able to capture, with good precision, the behavior and magnitude of the FWI in both regions, as was demonstrated by comparing the results with the benchmarks of [23,30], which were based on several other studies (see references therein). The model also performed better than other models in the literature, although it is important to note that there were not many studies that developed fire forecasts for *Colombia*. Additionally, the model was able to identify large values of the FWI except in the center of *Magdalena*, avoiding false alarms in most of the domain. The uncertainties seemed to be related to the changes in the wind field produced by its topography. Nevertheless, the model had good precision and the land cover analysis reduced the uncertainty by using this measurement as a weighted input to perform the calculation of the probability of the occurrence of a wildfire.

#### 3.1.2. WRF Bias Correction Technique Evaluation

Although the previous model reproduced the meteorological conditions that serve as input to calculate the FWI fairly well, the WRF-1D was included for three reasons: (i) this bias correction technique is new (at least to the extent of our knowledge) for meteorological variables, since it has only been used for air quality [19,20]; (ii) it is important to compare the bias correction technique with a pure model to decide whether it can be combined with the 3D models to have a good spatial coverage, and also to reduce errors in the pixels that contain the stations; (iii) evaluating the drawbacks of this technique allows us to address and document them for future investigations, especially considering that the results are very encouraging.

The WRF-1D model highly increased the precision of the WRF model, for all the variables except for precipitation (Figure 5). Before describing the results, it is important to mention that all the calculations were made for each station with three-hourly data. Thereafter, the statistical values were averaged. First, the improvements produced by the bias technique must be described, as they are used to calculate the FWI, and because this technique has been rarely, or never, used for meteorology. This will also allow us to understand the drawbacks of the model and to trace a path for future research.

Indeed, the WRF-1D model was able to highly increase the accuracy of the WRF model to reproduce the relative humidity and 2m air temperature. Its statistical parameters were excellent when compared with Guevara-Luna et al.’s [30] benchmarks for these two variables. In terms of wind speed, the results were good and borderline excellent in most of the stations. In fact, in some cases the correction managed to change the statistic from good to excellent (e.g., Appendix A in Figure 5). Another interesting feature is that the model was excellent at reducing the magnitude bias but was not as good at improving the representation of trends. The former can be observed when comparing the Rho and the RMSE values. On the other hand, the caveat of the model is the precipitation. Although the model was able to improve its results, its statistics for Rho and IOA oscillated at the 0.5 mark (good to bad category in [30]) and between four to seven (good to bad category in [30]) in the RMSE, which shows a systematic overestimation of the model when compared with the gauge precipitation values. Many reasons could explain this phenomenon, and here we mention two that are important: (i) the gauge coverage has to be improved, since one model pixel only contains one or two stations, and hence this comparison must be made with caution; and (ii) *Cundinamarca* is located in a highly complex terrain, and its precipitation events tend to be related to local convective regimes (e.g., [64,65]), which need a high spatial resolution (in the grid and in the topography as shown by [30]) to improve the representation of convection and, in turn, the magnitude and behavior of the precipitation. 

These results show that a bias correction technique based on neural networks managed to highly improve the prediction capability of the WRF model for every evaluated variable. Nevertheless, they also show the need to further investigate the correction technique in relation to precipitation, for example, by applying it in a place with a larger number of gauges and without complex terrain. This would allow us to better evaluate and understand the limitations and advantages of the technique in order to improve it.

With the meteorological results validated, the FWI was calculated and evaluated following the exact same procedure as for the 3D neural net models. The FWI magnitudes and trends were better described when calculated using the bias-corrected meteorology, compared with the raw WRF data (Figure 6). However, the results did not improve as much as expected, since precipitation is a vital part of the FWI calculations. Nonetheless, the results improved enough to surpass the lower boundary of the good benchmark. To show that a large percentage of the uncertainty was related to the precipitation, we repeated the calculations using all the corrected meteorological variables, but replaced the corrected precipitation with the observed one. Through this procedure we found that the model was excellent in the statistical and in the categorical parameters for five of the six stations evaluated, since the S4 station had some errors possibly related to the wind. This proves that the technique can greatly improve the precision of the FWI forecast and also that big efforts have to be made in the future to reduce the uncertainties related to precipitation, something beyond the scope of this paper. 

On the other hand, we also included a bias correction directly for the FWI. For this, we used the structure of Figure 3, with the addition of one more dense network before the output, since the FWI was calculated for a daily temporal resolution. As observed in Figure 6, the error related to the correction of the FWI fell between the FWI calculated from the full meteorological correction and the FWI calculated with the observed precipitation. In fact, every statistic was between good and excellent for the benchmarks reported by [30]. Additionally, the model did not produce a significant number of false alarms (0.4) and managed to identify more than half of the events in most stations. Consequently, the POC showed that the model worked with good precision more than 60% of the time. This shows that both approaches had good results and could be used to reproduce and predict the FWI in *Cundinamarca*. 

### 3.2. Occurrence Probability of a Wildfire

After the meteorological models were validated, the FLCI calculation had to be evaluated, as described in Section 2. *Cundinamarca* and *Magdalena* had an accuracy rating of 0.74 ± 0.012 and 0.96 ± 0.008, respectively. As mentioned before, the model’s predictive capacity was assessed using the ROC curve (Figure 7) to plot the sensitivity as a function on specificity for a binary classifier system, as suggested by [39]. In this curve, a value of 0.5 indicates that the prediction is as good as a random estimation, while higher values indicate better predictions. *Magdalena’s* FLCI area under the ROC curve (AUC) was close to one, statistically proving that the model can reliably predict forest fires in this area. *Cundinamarca’s* FLCI, on the other hand, showed a different sensitivity and specificity analysis, as the sensitivity ranged from 0.57 to 1 and the AUC was 0.79. Despite these different results, in both cases the model is considered as a good predictor of wildfires in the study areas [40].

In fact, as shown in Figure 8, the networks produced similar weights to what was expected from the literature (e.g., [9,39,45]). Furthermore, the results showed a clear and important difference between the relevance of the variables in *Cundinamarca* from those in *Magdalena*. In *Cundinamarca*, the NDVI marked the highest relevance, followed by the NDWI and LCT, respectively. As for *Magdalena*, the NDWI was the most relevant variable, followed by the NDVI and LCT. These results were aligned with Armenteras et al.’s [2] findings of a significant positive correlation between burned area and rainfall anomaly for the region where *Magdalena* is located, but not for the region where *Cundinamarca* is.

The Bayesian model outputs were compared with the outcomes of a deterministic first order model, as described in Section 2. Although not shown here, the relative error between both model’s results was calculated. In the case of *Magdalena*, the relative error between the two models’ results was minimal when the NDWI weight was greater than 0.7, and the LCT was greater than the NDVI. This was expected since it matched the findings of several authors that have studied wildfire prediction (e.g., [9,39,45]). However, for *Cundinamarca*, the relative error between the Bayesian model results and the deterministic model was minimal when the LCT was greater than 0.5 and the NDVI was lower than 0.1. These results also coincide with the relative relevance results presented in Figure 8.

Some conditions were considered as potential explanations of the differences among both case studies’ modeling results. In *Magdalena*, wildfires had a clearer seasonal variability, with patterns in frequencies and intensity. In *Cundinamarca*, in contrast, fire events distribution tended to be more randomized (not shown). Thus, during the Bayesian model learning phase, the *Magdalena* model could have been receiving “more regular” information than *Cundinamarca’s* model. Another potential reason for these differences is the inability to include reliable ignition data, which are largely produced by human intervention, so the model was not able to include this important parameter. This could have led to failures in the probabilistic prediction, especially in *Cundinamarca*, where most of the territory was severely intervened. 

### 3.3. Global Vulnerability

As explained in Section 2, global vulnerability to fires comprises ecological and socioeconomic vulnerability. Figure 9 shows the global vulnerability results obtained for *Cundinamarca* and *Magdalena* for 27 December 2020 as an example. Both maps show dissimilar particularities, explained by the dichotomic analysis. On the one hand, *Magdalena’s* global vulnerability map had higher conditions of vulnerability in urban areas. This situation can be explained by a poor capacity of response to wildfire events, and by the high presence of green areas within urban areas. On the other hand, most representative urban areas in *Cundinamarca* did not have a global vulnerability category assigned, which can be interpreted as a non-existing vulnerability to wildfires. This condition can relate to a high response capacity to wildfire events and to a higher density of urban infrastructure.

In general, the *Cundinamarca* and *Magdalena* global vulnerability maps underpinned the importance of assessing ecological vulnerability, as most areas categorized as moderate and high corresponded to ecosystems classified as “endangered” and “critically endangered” due to human intervention [49], or due to having land covers with a higher susceptibility to combustion. This classification, in turn, can lead to new policy and management approaches that prioritize these areas. Likewise, some processes such as changes in land use, changes in vegetation cover and climate change are recognized as favoring wildfires. 

In relation to the socioeconomic vulnerability assessment, the implementation of the WUI required some adjustments due to urban patterns. In *Colombia*, the populated centers and the distance between them forced a reduction in the diameter of the WUI, as the adjoining urban areas generate continuity. It is important to consider this aspect when analyzing vulnerability and response capacity in small urban and peri-urban highly populated areas (Figure 9). At finer scales, a global vulnerability analysis is more dynamic, as it shows a higher variability of features and the factors they arise from over space. In contrast, finer scales in global vulnerability maps facilitate the identification of other aspects related to social groups and smaller communities. In terms of time scales of information, in *Colombia*, most socioeconomic data are measured in five-year periods, reducing the capacity for a vulnerability assessment on behalf of institutions and local authorities.

As the proposed global vulnerability assessment comprises the analysis of eight criteria, and considering the level of urbanization and human intervention in *Cundinamarca* and *Magdalena*, the spatial overlapping of criteria should be considered during risk management decision-making processes. 

### 3.4. Risk Management

#### 3.4.1. Risk Daily Maps for the Study Areas

The integration of the global vulnerability criteria and the probability of wildfire occurrence leads to the identification of fire risk. This analysis and its subsequent proposals for different strategies were designed as general as possible to be adapted to different scenarios. However, as an example, the fire risk was modeled and mapped for 27 December 2020 (a very critical day in the study period) for both study areas. Figure 10B–D shows the probability of wildfire occurrence in *Cundinamarca* and *Magdalena* for 27 December 2020. The Wildfire Risk Map was also constructed for the same date for the two areas (Figure 10A–C) in a finer spatial scale.

As observed in the map (Figure 10B), the probability of fire occurrence was mostly very low and low. However, as the results in the risk map (Figure 10A) considered the socioeconomic and ecologic vulnerability, the areas under risk were slightly different than those areas with higher categories of probability of wildfire occurrence. This situation can be observed in the southern and central–eastern areas of *Cundinamarca* where very low probabilities of wildfire occurrence were predicted, but low categories of risk were estimated. In *Magdalena,* the resulting maps (Figure 10C,D) showed a different situation. In contrast to *Cundinamarca*’s risk map, *Magdalena* showed higher categories of risk in those suburban and urban areas. This condition may have been due to a reduced response capacity and the presence of very vulnerable land covers. These results highlight the importance of response capacity and land cover vulnerability in order to assess risk.

These predicted and validated results (Figure 10) demonstrate the advantages of this EAS model: it delivers only one global risk value, and it acts immediately upon factors identified 24 h prior, efficiently modeling meteorological variables and land cover. Locally analyzing high fire probability and fire risks allows guided action, focused on reducing vulnerability. Likewise, it can aid in the assessment and identification of major potential harm. This can, in turn, have positive repercussions in terms of reducing the incidence of fires. 

As explained, a forest fire can develop in rural, urban and periurban areas that are prone to risk. This risk has to be systematically managed in terms of reducing uncertainty and implementing preventive policies to anticipate and mitigate disasters. However, in general, current risk management is fragmented due to political and administrative boundaries that are not capable of incorporating the response capacity of towns towards strategic or long-term planning [66]. Indeed, at national and regional levels, government risk management agencies are in charge of training and directing. In contrast, at a local level, authorities can manage the risk through integral, strategic and participatory decision making. To design and implement these measures, coordination between jurisdictions must be promoted, avoiding political conflicts between decision makers at the various levels and fostering efficiency in risk response by making timely decisions regarding mitigation strategies. These measures will be further described in the next two sections. 

#### 3.4.2. Early Alert Actions

The information on the probability of wildfire occurrence and vulnerability to forest fires is the basis for the creation of risk maps and EASs. Consequently, decision makers can use this information to design forest risk management policies; specifically, they can build fire extinction plans and spatialization maps in the medium and long term. To design geographical information system tools with machine learning, fire probability maps were created. These include risk based on the frequency rate, and were, in turn, combined with hazard maps created from coverage maps and the fuel potential of vegetation. These results can aid in the prevention of and reduction in forest fires, through the design and implementation of fuel management and territory conditioning measures [67].

Moreover, the participation of the communities in these processes is relevant, as it is the basis for a coregulated management between the community and territorial authorities. Hence, this participation adds value to the processes of the planning, execution and monitoring of restoration in the areas affected by wildfires. The proposed protocol defined three levels: (i) Red alert: To take action, warning the disaster prevention and assistance systems of the wildfire with adverse effects on the population. This requires immediate attention on behalf of the community and assistance to stakeholders. An alert is issued only when the identification of an extraordinary event indicates the probability of an imminent wildfire event and when the severity of the phenomenon entails the mobilization of people and equipment; (ii) Orange alert: To prepare, indicating the presence of a possible phenomenon. It does not imply an immediate event and, as such, it is a message towards information and preparedness. The warning implies continuous monitoring, as the conditions can lead to the development of an event, although without being required to remain alert; (iii) Yellow alert: To be informed, an official message is provided that explains where relevant information is publicly disclosed. Therefore, observed, reported or recorded events are required to be informed and the message may contain some forecast elements as a guide. It is different from the warning and the alert, as it is only intended to inform.

This protocol is activated after the competent technical authority in charge of assessing risks of forest fire events evaluates the situation, and its recommendations must be followed by public (political) authorities depending on the level of alert (Table 3). It is important to mention that if the alert materializes into a fire 24 h later, authorities should investigate the area to find possible causes of the fire and activate a protocol to mitigate damage and extinguish the fire.

Fire management programs must be implemented through integrated prevention planning, which involves matters related to environmental education, local and regional campaigns through different media, personnel training, equipment supply, implementation of a fire detection system, implementation of communication systems, fuel management and fire suppression. The fire management program is dynamic in nature. Thus, it must be periodically adapted because as institutional and social participation increases, more knowledge is gathered, and the matrix, created according to the use of land in the surroundings of the conservation unit, adopts alternative measures to guarantee the sustainability of production, avoiding the misuse of fire [68].

#### 3.4.3. Prevention and Mitigation Strategies

To define the Soil Management Practices (SMP), different public policy documents on fire management in Latin American countries were analyzed. As a result, these SMPs (Table 4) were derived from the most common practices, and consequently, most of them have some of these policies already in place, have very close examples or are institutionally ready to implement them. SMP are categorized in before- and after-wildfire event practices. As before-wildfire SMP aims to diminish the global vulnerability on the territories, they can be regulatory, educational and management strategies to change land cover (SMP 1.1, 1.2 and 1.3). These strategies consider continuous changes in land cover uses and practices as catalyzers of wildfires, as they increase combustible loads [69]. On the other hand, after-wildfire SMP targets to protect soils after the event has happened [70] (SMP 2.2 and 2.3), and they must be conducted before the rainy season [71]. 

Considering the variety and severity of the impacts of a wildfire event on ecosystems, such as the promotion of erosive conditions and the alterations in hydrological and nutrient dynamics (e.g., [72]), a set of long-term SMPs are proposed (SMP 2.1) to recover and reforest burned areas. This process under natural conditions can take between 3 to 15 years (e.g., [73,74]) depending on the type of original land cover succession times. To be more precise and produce a more complete evaluation, in Appendix A we present a susceptibility analysis and show what SMP actions could be applied depending on the land cover gradient and its susceptibility.

## 4. Conclusions and Future Work

Despite a wildfire event resulting from the interaction of several elements, which includes a random ignition event, the proposed method supplies local decision makers with useful and practical information to prevent the event, or to reduce its impacts. In this research, we developed machine learning and remote sensing technologies to offer tools for detecting and monitoring wildfires. These technologies were integrated with risk assessment and global vulnerability factors to generate information for the preventive risk management of wildfire events. The prediction of meteorological variables, enabled by neural networks, alongside the analysis of coverage registered by satellites, led to the identification of the probability of forest fire occurrence. These technological results can blend with an understanding of the vulnerability of vegetal covers in urban interface areas and the response capacity of each territory, enabling efficient and informed decision making towards the optimization of risk management. 

Models for wildfire prediction, thus far, are unable to incorporate the ignition on its quantitative approach. Hence, wildfire prediction is not related to ultimate events, but has a high potential to feed short-term decisions in the local territories, considering the time and space scale of its results. In this paper, a meteorological model and a model using land cover are designed. The meteorological model incorporates a 3D neural network, a bias correction WRF-1D, combined with the Naïve-Bayes classifier to predict the probability of wildfire occurrence up to 24 h ahead. 

The prediction relevance distribution of land cover variables was different among case studies. These results support academic researchers that argue in favor of differences in the conditions that facilitate wildfire occurrence in every region. To prevent wildfire events or mitigate its impacts, relevant and integrated information is key. Thus, the proposed EAS considers vulnerability, the probability of wildfire occurrence and risk maps. Each of these factors has complementary information that can facilitate the design and implementation of strategies and the use of tools in the territories. 

A risk management approach must be preventive and include trustworthy and cost-efficient risk mitigation measures. To achieve this, five types of risk areas were categorized (very low, low, moderate, high, very high/extreme). These can be the basis for risk management strategies, according to the capabilities of each territory in terms of disaster risk reduction, preparedness for response and cooperation between government authorities, local agencies and private sectors to face and reduce the impact of wildfires. Likewise, to improve the fire early alert system, important criteria must be considered such as crop type, access routes or communication webs in the area, which can aid in an enhanced prompt response. Another issue encountered when assessing the risk of fire is the difference between spatial resolution levels of vegetation cover maps and socioeconomic vulnerability data, which hampers the evaluation of the risk at a local scale, by causing a generalization in the fire risk level. 

Regarding future research, there are many potential paths to be followed. Here, we describe some of them for further investigation. As pointed out by one reviewer, one path could consist of adding an emission forecasting tool to the EAS. In this sense, the model would not only be capable of identifying possible events but would also be able to forecast the amount of emissions (of, for example, -CO_2_ or particulate matter) generated by a wildfire. This could be further validated by wireless sensors. Another research path could be to develop an ML learning tool combined with Monte Carlo simulations to directly quantify the probability of a wildfire. This ML tool could also be used to perform sensitivity experiments, which could aid in the evaluation of policies before their implementation. Another interesting road would entail applying the EAS described in this paper to the Amazon Forest, and add, for example, an evaluation of forest fragmentation due to wildfires. These research paths could, in turn, lead to future important and exciting investigations related to the EAS, ML tools and policy.

## Figures and Tables

**Figure 1 sensors-22-08790-f001:**
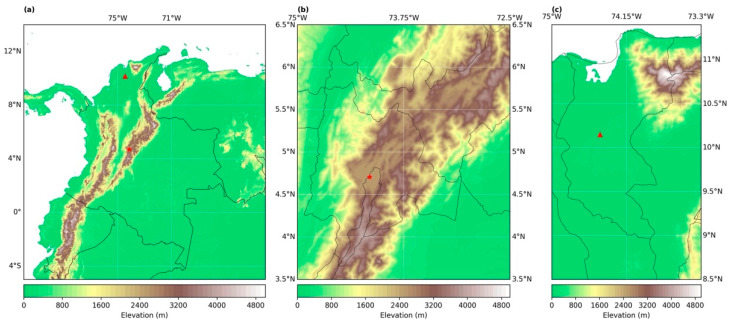
(**a**) Location of *Colombia*, with *Cundinamarca-Bogotá* (star) and *Magdalena* (triangle) and its surrounding topography. (**b**) Close up of the surroundings of the *Cundinamarca-Bogotá* departments. (**c**) Close up of the surroundings of the *Magdalena* Department. The black lines of panels b and c represent the political division of the country.

**Figure 2 sensors-22-08790-f002:**
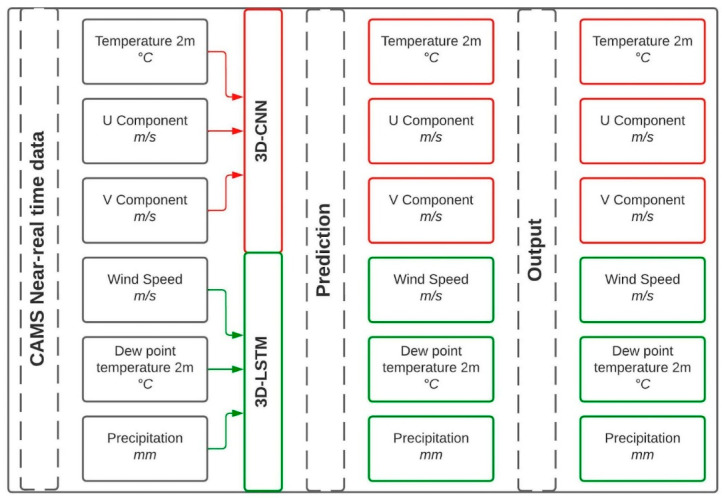
Flowchart of the inputs, models and outputs of the 3D neural networks. Notice that each model focuses on three variables due to their forecasting performance. This is denoted with the arrow colors and the edge color of the boxes.

**Figure 3 sensors-22-08790-f003:**
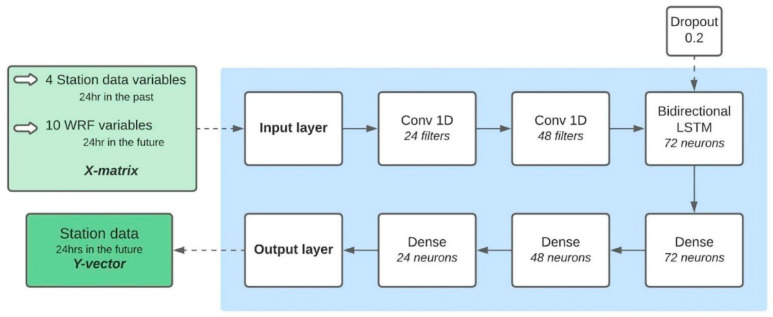
WRF-1D neural network structure. Notice that every box has the number of neurons/filters in the layer and that the dropout percentage of a layer is also indicated. The light green box shows the input variables (X-matrix) and the green box indicates the output (Y-vector).

**Figure 4 sensors-22-08790-f004:**
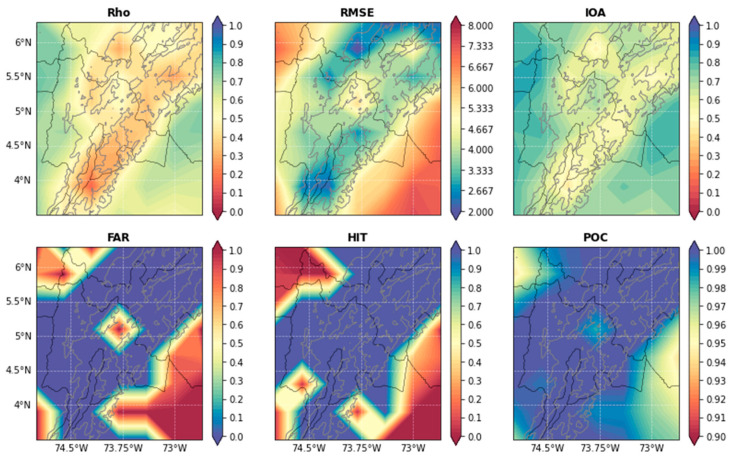
Statistical parameters of the neural net for *Cundinamarca.* Notice that the title of each panel represents the parameter calculated, and that the panels have different color bars. The gray contours indicate the *Andean* Mountain ranges.

**Figure 5 sensors-22-08790-f005:**
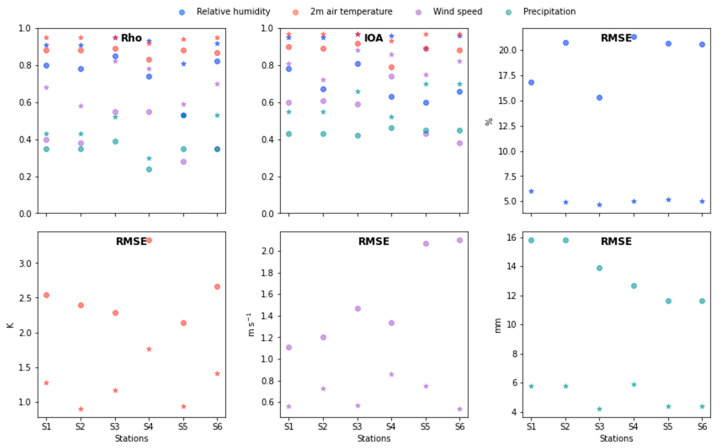
Statistical evaluation of the model performance for relative humidity, 2m air temperature, wind speed and precipitation for the selected 160 days (Section 2). The circles represent the results before the model was bias corrected and the stars show the results after the correction was applied. Notice that the RMSEs have different units depending on the variable.

**Figure 6 sensors-22-08790-f006:**
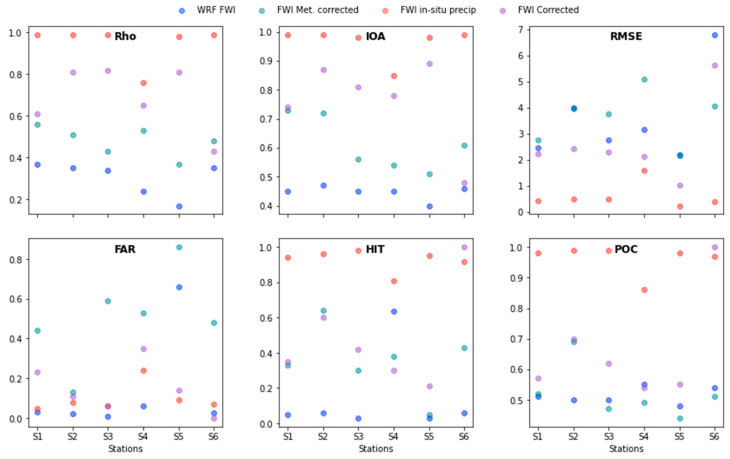
Statistical evaluation of the model performance for the FWI for the selected 160 days (Section 2). The blue circles represent the statistics of the WRF model before any correction. Dark cyan circles show the FWI calculated with the meteorological corrected variables. Red circles indicate the FWI calculated with the meteorological corrected variables and the observed precipitation. Purple circles show the FWI directly corrected by the WRF-1D.

**Figure 7 sensors-22-08790-f007:**
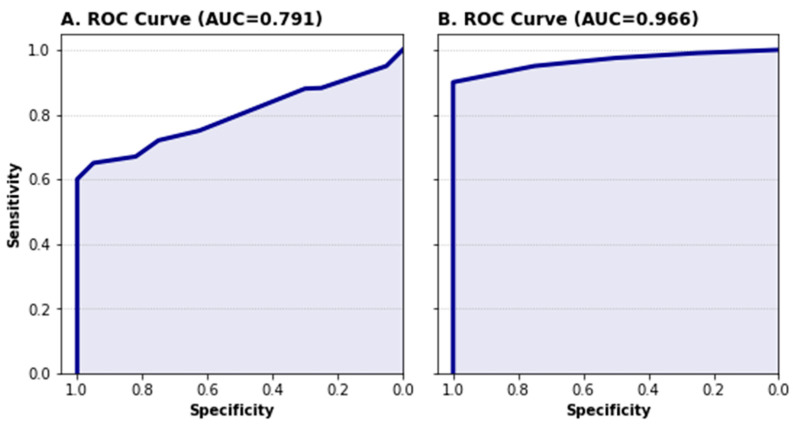
ROC curve analysis of (**A**) *Cundinamarca’s* FLCI and (**B**) *Magdalena’s* FLCI, respectively.

**Figure 8 sensors-22-08790-f008:**
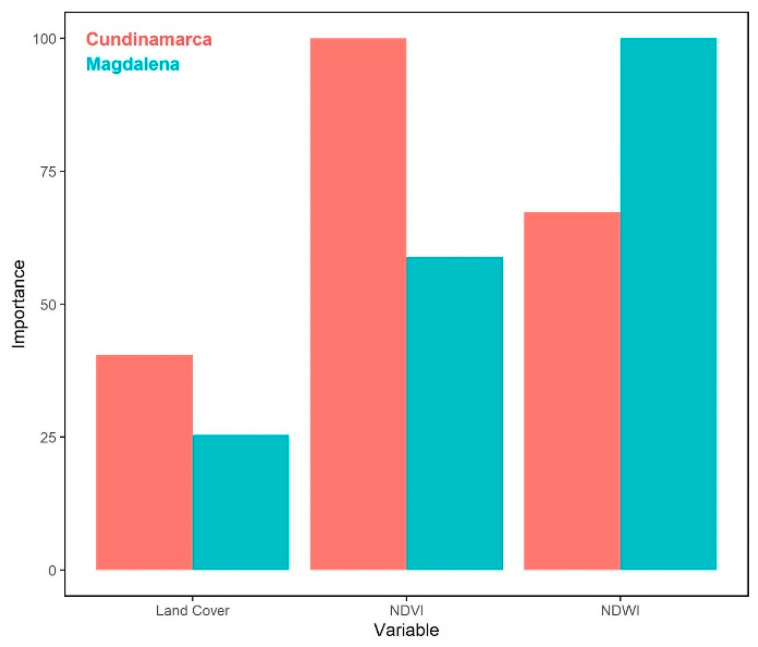
Degree importance for each feature in the X-matrix of the Naïve-Bayes classification. Red bars represent *Cundinamarca*, and the blue bars represent *Magdalena*.

**Figure 9 sensors-22-08790-f009:**
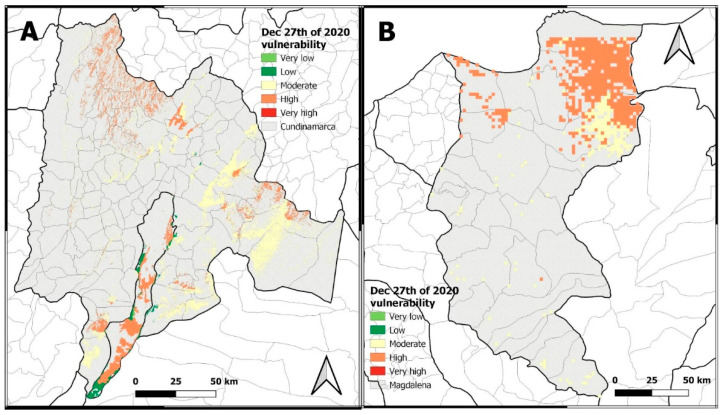
Global vulnerability of *Cundinamarca* (**A**) and *Magdalena* (**B**). Colors represent the global vulnerability category as very high (red), high (orange), moderate (yellow), low (green), very low (light green).

**Figure 10 sensors-22-08790-f010:**
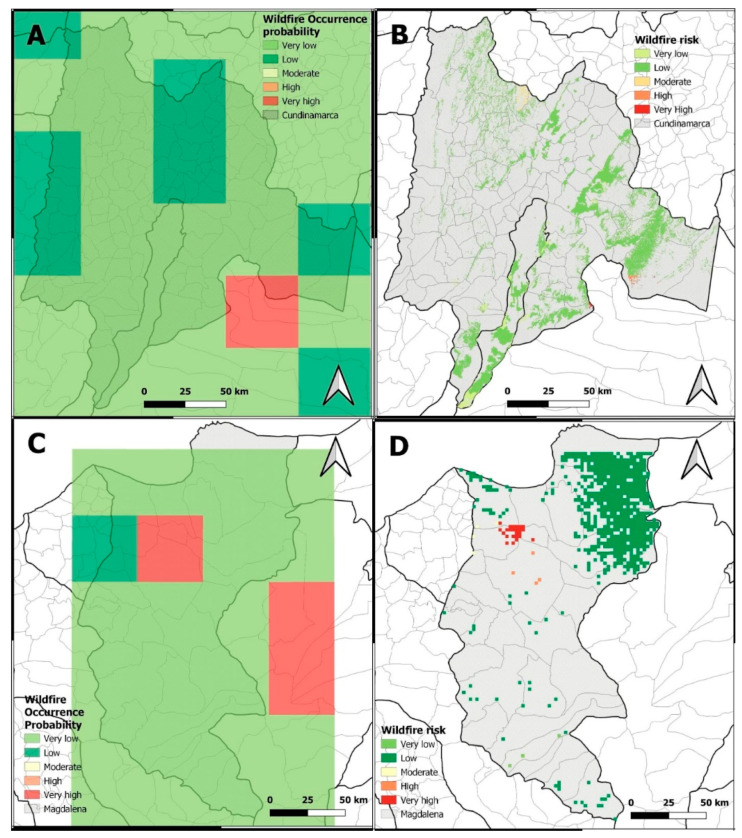
Wildfire Occurrence Probability of *Cundinamarca* (**A**) and *Magdalena* (**C**); Wildfire Risk Map in *Cundinamarca* (**B**) and *Magdalena* (**D**) on 27 December 2020. Colors represented as very high (red), high (orange), moderate (yellow), low (green), very low (light green).

**Table 1 sensors-22-08790-t001:** Coverage classification and fuel type.

Predominant Type of Coverage	Fuel Type	Susceptibility Category	Value
Forest	Shrubbery	Low	2
Fragmented Forest	Trees	Medium	3
Gallery and riparian forest	Trees	Low	2
Shrubland	Shrubbery	High	4
Mosaic of crops, pastures and natural spaces	Grass/Herbs	Very High	5
Mosaic of pastures with natural spaces	Grass/Herbs	Very High	5
Mosaic of pastures and crops	Grass/Herbs	Very High	5
Crop Mosaic	Herbs	High	4
Grass	Grass	Very High	7
Grassland	Herbs	Very High	6
Glacial and snowy areas	No combustible	-	1
Urban areas	No combustible	-	1

**Table 2 sensors-22-08790-t002:** Fire sensitivity classification for each index for risk assessment.

Category	FWI	Normalized FWI	WFLC	WFP	Global Vulnerability	Fire Risk
Very Low	<5.2	0. 052	0–0.2	0–0.126	1–3.8	0–0.48
Low	5.2–11.2	0.052–0.11	0.2–0.4	0.126–0.255	3.8–6.6	0.48–1.68
Moderate	11.2–21.3	0.11–0.213	0.4–0.6	0.255–0.405	6.6–9.4	1.68–3.81
High	21.3–38	0.213–0.38	0.6–0.8	0.405–0.59	9.4–12.2	3.81–7.19
Very High/ Extreme	38–100	0.38–1	0.8–1	0.59–1	12.2–15	7.19–15

**Table 3 sensors-22-08790-t003:** General Protocol for public authorities in the case of a fire alert 24 h prior.

Alert Level	General Recommendations for Public Authorities	Towns with Low Risk	Towns with Medium Risk	Towns with High Risk
Yellow Alert	Scan the area of alert for flammable objects (e.g., debris) or sparkling objects (e.g., lit cigarettes, lighters, fireworks) and remove them. Suspend campfires and other fire-related activities in the area and ridge the campsite with rocks or dirt. Water the grass around the area.	Identify and characterize the area and forest to be protected (risk map). Educate and train fire departments and other relief agencies that have forestry brigades.	Identify and provide the necessary tools, equipment and access. Have a historical fire log of events. Activate the environmental surveillance network. Review and verify the capacity of response agencies.	Verify the status and availability of resources. Identify water sources or storage tanks. Have an up-to-date risk map.
Orange Alert	Scan the area of alert for flammable objects (e.g., debris) or sparkling objects (e.g., lit cigarettes, lighters, fireworks) and remove them. Suspend campfires and other fire-related activities in the area and ridge the campsite with rocks or dirt. Water the grass around the area. Prepare and bring fire equipment and human resources to the area (e.g., firefighters, fire engines, extinguishers, protective gear).	Activate a response network in charge of monitoring the risk of fires in the area.	Verify the tools, equipment and accessories necessary for care. Prepare hydrometeorological reports to ascertain the behavior of the climate.	Activate the sound and/or visible siren (beacons) of vehicles.
Red Alert	A timely (in the shortest possible time) and effective (location-based) warning related to the observation of columns of smoke or sources of fire, which can cause forest fires. Scan the area of alert for flammable objects (e.g., debris) or sparkling objects (e.g., lit cigarettes, lighters, fireworks) and remove them. Suspend any human activities and evacuate the area. Water the grass around the area. Prepare and bring fire equipment and human resources to the area (e.g., firefighters, fire engines, extinguishers, protective gear). Remove any vehicles or heavy machinery and tools that can cause a spark, produce heat or contain flammable liquids (e.g., fuel). Alert neighboring political authorities for a possible activation of their contingency plan, which can include greater efforts to prepare for a larger event (helicopters, international aid, among others).	Activate and mobilize resources of authorities responsible for control, extinction, liquidation and recovery.	Establish functional entry, exit and/or evacuation routes.	Prepare the line of defense or backfire to have a unit in each section that verifies that the possible fire does not exceed it.

**Table 4 sensors-22-08790-t004:** List and definition of Soil Management Practices (SMP) chosen for before and after a fire event. Prefire event practices are management and public-policy type, while postfire practices are management type only.

Type of Practice	Soil Management Practice	Definition
1. Practices before a wildfire	1.1. Mechanical reduction, modification, use and/or elimination of forest fuels.	Modification of land covers susceptible to wildfires in order to reduce the risk of ignition.
	1.2. Education and training focused on fire management on the soil	Community training related to wildfire management.
	1.3. Regulatory actions and government management.	Strengthening regulatory system actions towards wildfire management.
2. Practices after a wildfire	2.1. Recovery of land cover in affected areas by wildfires	Changes towards land cover not susceptible to wildfires in affected areas.
	2.2. Sanitation on forest masses	Cleaning affected areas, eliminating fuel forest masses.
	2.3. Postfire soil control	Actions to reduce susceptibility to erosion in affected areas.

## Data Availability

All the data and scripts will be made available on request to the authors.

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
