# Peer review of "Design of a Forest Fire Early Alert System through a Deep 3D-CNN Structure and a WRF-CNN Bias Correction"

_sensors, 2022, doi:10.3390/s22228790_

Round 1
Reviewer 1 Report
This is an interesting paper that worth’s publication and deals with an early detection system for vulnerability prediction and not real time identification. This approach is very important for risk management and the authors provide also a scheme.
Since they employ prediction of meteorological data they could examine if there is chaotic behavior for the time horizon of 7 days they employ and use appropriate algorithm modification (see for example https://doi.org/10.1007/s00521-021-06266-2.
Could the framework be combined with real time identification of several quantities such as -Co2 concentration measured through wireless sensors?
Author Response
First of all, we thank the reviewers for their valuable comments, which have helped us to greatly improve the manuscript. In the next lines, we answered the reviewers’ comments. We include our answers in blue color, so that the differences can be easily identified and tracked.
Reviewer 1:
This is an interesting paper that worth’s publication and deals with an early detection system for vulnerability prediction and not real time identification. This approach is very important for risk management and the authors provide also a scheme.
We thank the reviewer for their comments and for their opinion about the paper. We will now show how we follow their suggestions.
- Since they employ prediction of meteorological data they could examine if there is chaotic behavior for the time horizon of 7 days they employ and use appropriate algorithm modification (see for example https://doi.org/10.1007/s00521-021-06266-2.
We thank the reviewer for their comment. We followed the procedure of the paper and found that the safe forecasting horizon is 8.3 days. Indeed, the system is chaotic, as we expected, since Lorenz (1963) showed that meteorological systems are highly chaotic. Additionally, we increased the forecasting horizon from 5 to 10 and found that after 8 days the errors start to strongly increase.
We added this to the results section in the main text. We also included Lorenz’ paper and the paper suggested by the reviewers. See below. Notice that the new references have their names, but in the main text they have numbers to facilitate the review.
“Before describing the model’s results, to reduce the uncertainty, we calculated the forecast horizon following the work of Stergiou and Karakasidis [35] (consult the reference for methodological details). The horizon is 8.33 days and the system was found to be chaotic. This was to be expected since Lorenz [63], in his seminal paper, showed the importance of chaotic systems, and used the meteorological system as an example. Moreover, we performed a battery of experiments (on the 3D networks), where we changed the forecasting horizon from 5 to 10 days. We found that, after 9 days, the errors start to sharply increase.”
2. Could the framework be combined with real time identification of several quantities such as -Co2 concentration measured through wireless sensors?
We greatly thank the reviewer for this comment. We think it is worth trying to follow this research path in the future. We reckon that it is completely possible to do it, but in a new paper, since it is out of the scope of this one. Nevertheless, we included this idea (and others) into the future research part, as was suggested by the second reviewer. See below our addition to the main text.
“Regarding future research, there are many potential paths to be followed. Here, we describe some of them for further investigation. As pointed out by one reviewer, one path could consist of adding an emission forecasting tool to the EAS. In this sense, the model would not only be capable of identifying possible events, but would also be able to forecast the amount of emissions (of, for example, -CO2 or particulate matter) generated by a wildfire. This could be further validated by wireless sensors. Another research path could be to develop a ML learning tool combined with Monte-Carlo simulations to directly quantify the probability of a wildfire. This ML tool could also be used to perform sensitivity experiments, which could aid in the evaluation of policies before their implementation. Another interesting road would entail applying the EAS described in this paper to the Amazon forest, and add, for example, an evaluation of forest fragmentation due to wildfires. These research paths could, in turn, lead to future important and exciting investigations related to EAS, ML tools and policy.”

Reviewer 2 Report
This manuscript proposed a novel forest fires early alert system based on deep learning, where 3D-CNN model was established for the task of interest. The performance of the proposed method has been validated based on experiments, with satisfactory results. The outcome of this research can be considered as an ideal solution in the real application. Overall, the topic of this research is interesting, and the manuscript was well organised and written. The detailed comments are provided as follows.
1. The contribution and innovation of the manuscript should be clarified clearly in abstract and introduction.
2. Broaden and update literature review on CNN and its application in resolving the real problems. E.g. Vision-based concrete crack detection using a hybrid framework considering noise effect.
3. The performance of deep learning model relies on the setting of hyperparameters. How did the authors set hyparameters of CNN in this research to achieve the optinal prediction performance?
4. It will be better to compare the proposed method with other existing methods in terms of forest fire assessment to validate the superiority of the proposed method.
5. More future research should be included in conclusion part.
Author Response
First of all, we thank the reviewers for their valuable comments, which have helped us to greatly improve the manuscript. In the next lines, we answered the reviewers’ comments. We include our answers in blue color, so that the differences can be easily identified and tracked.
This manuscript proposed a novel forest fires early alert system based on deep learning, where 3D-CNN model was established for the task of interest. The performance of the proposed method has been validated based on experiments, with satisfactory results. The outcome of this research can be considered as an ideal solution in the real application. Overall, the topic of this research is interesting, and the manuscript was well organised and written. The detailed comments are provided as follows.
We thank the reviewer for their comments. We hope that the paper can become the basis to new policies and be used in many countries as an alternative to their EAS, especially in countries that, like Colombia, don't have strong computing power. Below we describe how we followed the reviewer's suggestions.
- The contribution and innovation of the manuscript should be clarified clearly in abstract and introduction.
Thanks to the reviewer for its comment. We included the following sentence at the end of the abstract: “In conclusion, this paper creates an EAS for wildfires, based on novel ML techniques and risk maps.“ Nevertheless, including more information is difficult even if we would completely change the abstract. Hence, we decided to state a strong and concrete idea in the abstract which is then more carefully explained at the end of the introduction. This new paragraph was added at the end of the introduction:
“In general, this paper presents the design of an early alert system, with some novel approaches explained below. i) we used a state-of-the-art bias-correction method for meteorology, based on ML techniques, which, to the extent of our knowledge, is one of the first to be applied in an area with complex terrain (i.e., Colombia). ii) this is one of the first efforts to use a Navier-Bayes classifier to determine the probability of occurrence of a wildfire, an approach validated with several methods from literature. iii) we merged this approach with a vulnerability analysis in risk maps, which allows for the dynamic assessment of the risk and the daily changes. Likewise, it provides insights on how and where to proceed to reduce the global risk. These insights are merged into a new protocol, which includes short- and long-term actions. iv) Today, the access to, and development of, information in Colombia is still mostly centralized, with some national governmental agencies working on generating maps and tools that aid decision-making in the territories on coarse space-time scales. Providing an easy-to-use tool, with dynamic results at appropriate spatial scales, contributes to strengthening territorial (communitarian) sovereignty, by handing control, and promoting knowledge and skills, to local levels.”
- Broaden and update literature review on CNN and its application in resolving the real problems. E.g. Vision-based concrete crack detection using a hybrid framework considering noise effect.
We followed this suggestion. We added the paper mentioned here, and also 3 more. Nevertheless, we want to mention that our focus was more on wildfires and related models, more than the CNN. We hope that with the inclusion of these four new references in the introduction, the paper is more complete.
“We decided to use ML models, which especially relate to convolution neural networks (CNN), due to the strong results they have shown in multiple science fields. The convolutional networks have been mainly used for visual processing. For example, Yu et al (2022) create a state-of-the-art CNN to identify cracks that develop on concrete structures. The precision of their networks was greater than 90%, which shows their capabilities in civil engineering, and their potential in many other fields and applications. One of those applications is shown in Javanmardi et al. (2022). They created a method which reduces the limitations of CNNs when captioning images. This has increased the attention on the advantages of CNNs when identifying and classifying objects, and it also sheds light regarding the capability of the network when forecasting. One interesting example relates to air quality, which has been diversely studied. For example, Kalajdjieski et al. (2020) show that CNNs are able to predict the amount of pollution from satellite images, without using raw time series. This is certainly of great utility for places without ground stations. Another example is the work of Kabir et al. (2022), which also used satellite images to determine the spatial distribution of pollution. These works, and many others, some of which are referenced in this paper, have built the basis for the application of CNNs to different environmental problems, which can be solved with satellite images and/or time-series of data, as the first part of this paper shows.”
- The performance of deep learning models relies on the setting of hyperparameters. How did the authors set hyperparameters of CNN in this research to achieve the optimal prediction performance?
We followed the reviewer's comment and more clearly specified the selection of the hyper parameters and structure. The new paragraph is presented below:
“Two network-structures were designed to account for the better results that each structure produces depending on the simulated variables (Fig. 2 and Tables S1 - S2). To reduce the uncertainty to the greatest extent possible, a grid search method (e.g., [24]), which consists of more than 350000 tests, was implemented to find the best hyper parameters for each network structure. Moreover, an early stopping method was applied to prevent overfitting [25]. It is also important to mention that 6 other network structures were tested, thereafter selecting the one producing the smallest errors.”
- It will be better to compare the proposed method with other existing methods in terms of forest fire assessment to validate the superiority of the proposed method.
We encountered some issues related to this comment. When we evaluated the model, we decided to follow some benchmarks and criteria because comparing the model with others is difficult, as there are not many papers that have tried to forecast wildfires in Colombia. Colombia has a very rough topography, which produces high changes in wind patterns and their conditions are not very comparable to other places. Hence, we decided to use benchmarks that we knew already existed. That said, we followed the reviewer's suggestion and compared our results with two papers that include Colombia in their analysis. This was done only for the 3D models, since these were then compared with the WRF-1D. From this analysis, it is clear that the results are more precise in this model. Nonetheless, this comparison between a time-series and satellite forced model has to be carefully made, since the nature and characteristics of both are really different. Here, we present what was added to the main text, and, again, we decided to leave the name of the reference to help the review process.
“To have a more robust evaluation of the 3D models, we compared our results with previous studies. For example, Anderson et al. (2021) developed an alert system for protected areas of South America. They calculated the HIT and FAR parameters and found that their model has good precision. However, the model presented herein has a 40% higher accuracy in the HIT category and produces 5% less false alarms. On the other hand, Di Giuseppe et al. (2016) used a forecast model to calculate the potential predictability of wildfires around the globe. They found a categorical score very close to ours. In fact, their score oscillates between 0.6 to 0.9, similar to the values in Fig. 4. Nevertheless, they forecasted the entire world, and not a specific zone. Hence, although the comparison is possible, it is important to bear in mind that this paper focuses on only one zone. Hence, their precision could increase more. These two comparisons show that 3D networks have a great accuracy when compared to other models and methods, and that, in contrast to other works, this model was combined with a land cover analysis, which improves the results. ”
We also included one phrase in the summary of that subsection: “The model also performs better than other models in the literature, although it is important to note that there are not many studies that developed fire forecasts for Colombia.”
- More future research should be included in the conclusion part.
We combined this comment with the second comment of the first reviewer. Below that comment, we included our addition to the main text. We think this greatly improves the document, especially as we reckon that new ideas are always welcome to advance these types of applications.

Reviewer 3 Report
This paper explores machine learning and remote sensing technologies to detect and monitor wildfires, in order to provide scientific evidence for further preventive risk management of wildfire events. Overall, the manuscript is very well-written with perfect logic, and can be accepted as long as the following minor issues are addressed:
1. The abbreviation ‘WRF’ is suggested to be expanded in the abstract when it first appears;
2. The reviewer understands that the bi-directional LSTM should perform better than ordinary LSTM, yet bi-directional LSTM is usually adopted based on the concept that it is necessary to process both positive- and reverse-order sequence data, e.g., the translation of sentences. Therefore, it is recommended that the motivation or the underlying logic of applying bi-directional LSTM be illustrated in the manuscript.
Author Response
First of all, we thank the reviewers for their valuable comments, which have helped us to greatly improve the manuscript. In the next lines, we answered the reviewers’ comments. We include our answers in blue color, so that the differences can be easily identified and tracked.
Reviewer 3:
This paper explores machine learning and remote sensing technologies to detect and monitor wildfires, in order to provide scientific evidence for further preventive risk management of wildfire events. Overall, the manuscript is very well-written with perfect logic, and can be accepted as long as the following minor issues are addressed:
We greatly thank the reviewer for their comments. We followed their suggestions, which helped improve our manuscript.
- The abbreviation ‘WRF’ is suggested to be expanded in the abstract when it first appears
Done
- The reviewer understands that the bi-directional LSTM should perform better than ordinary LSTM, yet bi-directional LSTM is usually adopted based on the concept that it is necessary to process both positive- and reverse-order sequence data, e.g., the translation of sentences. Therefore, it is recommended that the motivation or the underlying logic of applying bi-directional LSTM be illustrated in the manuscript.
We thank the reviewer for this useful suggestion. We included the two reasons for the use of a bidirectional LSTM. We also added a citation (which is referenced here by names, but in the main text by a number, in order to help the reviewer find the reference) to back up the two-ways propagation. We added below our addition to the text. In the past, we have already used both LSTM and Bidirectional LSTM and found that the peaks are better represented in the bidirectional network, possibly due to the reverse-order sequence, which is capable of catching those steep increments.
“There are two main reasons behind the use of this structure: i) we tested 6 different structures with different combinations of layers (e.g., with and without bidirectional networks), and found that this configuration produces the smallest errors. ii) The bidirectional networks are able to produce nowcasting when used on time-series of data [21], due to their ability to reverse-order the data. This helps them identify strong events with more ease than, for example, a normal LSTM [35]. In fact, one of the drawbacks of the LSTM is that in some cases it is not able to reproduce the peaks of the variable they are forecasting (e.g., PM2.5), something that is largely reduced with the bidirectional network (see [21] and [23] for examples).”
